# Identification of Iguania Ancestral Syntenic Blocks and Putative Sex Chromosomes in the Veiled Chameleon (*Chamaeleo calyptratus*, Chamaeleonidae, Iguania)

**DOI:** 10.3390/ijms232415838

**Published:** 2022-12-13

**Authors:** Katerina V. Tishakova, Dmitry Yu. Prokopov, Guzel I. Davletshina, Alexander V. Rumyantsev, Patricia C. M. O’Brien, Malcolm A. Ferguson-Smith, Massimo Giovannotti, Artem P. Lisachov, Vladimir A. Trifonov

**Affiliations:** 1Institute of Molecular and Cellular Biology, Russian Academy of Sciences, Siberian Branch, Novosibirsk 630090, Russia; 2Department of Natural Sciences, Novosibirsk State University, Novosibirsk 630090, Russia; 3Institute of Cytology and Genetics, Russian Academy of Sciences, Siberian Branch, Novosibirsk 630090, Russia; 4Cambridge Resource Centre for Comparative Genomics, Department of Veterinary Medicine, University of Cambridge, Cambridge CB3 0ES, UK; 5Dipartimento di Scienze della Vita e dell’Ambiente, Università Politecnica delle Marche, 60121 Ancona, Italy; 6Animal Genomics and Bioresource Research Unit (AGB Research Unit), Faculty of Science, Kasetsart University, 50 Ngamwongwan, Chatuchak, Bangkok 10900, Thailand

**Keywords:** reptilia, karyotype evolution, chromosome fusion, repetitive elements, Squamata, microchromosomes

## Abstract

The veiled chameleon (*Chamaeleo calyptratus*) is a typical member of the family Chamaeleonidae and a promising object for comparative cytogenetics and genomics. The karyotype of *C. calyptratus* differs from the putative ancestral chameleon karyotype (2n = 36) due to a smaller chromosome number (2n = 24) resulting from multiple chromosome fusions. The homomorphic sex chromosomes of an XX/XY system were described recently using male-specific RADseq markers. However, the chromosomal pair carrying these markers was not identified. Here we obtained chromosome-specific DNA libraries of *C. calyptratus* by chromosome flow sorting that were assigned by FISH and sequenced. Sequence comparison with three squamate reptiles reference genomes revealed the ancestral syntenic regions in the *C. calyptratus* chromosomes. We demonstrated that reducing the chromosome number in the *C. calyptratus* karyotype occurred through two fusions between microchromosomes and four fusions between micro-and macrochromosomes. PCR-assisted mapping of a previously described Y-specific marker indicates that chromosome 5 may be the sex chromosome pair. One of the chromosome 5 conserved synteny blocks shares homology with the ancestral pleurodont X chromosome, assuming parallelism in the evolution of sex chromosomes from two basal Iguania clades (pleurodonts and acrodonts). The comparative chromosome map produced here can serve as the foundation for future genome assembly of chameleons and vertebrate-wide comparative genomic studies.

## 1. Introduction

The infraorder Iguania is one of the largest and most diverse groups of non-snake squamates, which includes two subclades: Pleurodonta (Iguanidae *sensu lato*) and Acrodonta (Agamidae and Chamaeleonidae), that diverged about 135–160 MYA [1,2,3]. The pleurodont clade is found in the New World and Madagascar, whereas the acrodont clade occurs in the Old World and Australia [4].

The family Chamaeleonidae consists of 12 genera [5]. All family members possess characteristic morphological traits (including the ability to change skin color, projectile tongues, prehensile tails, zygodactylous feet, etc.) [6,7]. The putative ancestral karyotype of Chamaeleonidae contains 6 pairs of metacentric macrochromosomes and 12 pairs of microchromosomes (2n = 36) [8]. A similar type of karyotype was described for numerous modern species of all the iguanian families [9]. Besides, some modern iguanian species possess derived karyotypes. For example, the chromosome numbers of chameleons range from 2n = 20 to 2n = 62, with most species showing chromosome number reduction via tandem and Robertsonian chromosome fusions [8].

Sex chromosomes have been found in only a few chameleon species. Female heterogamety was reported for the Madagascar chameleons of the genus *Furcifer*. Most species studied have heteromorphic ZW-type sex chromosomes with a large heterochromatic W or multiple Z_1_Z_1_Z_2_Z_2_/Z_1_Z_2_W systems [10,11]. Despite the lack of differentiated sex chromosomes, genetic sex determination (GSD) was demonstrated for the veiled chameleon (*Chamaeleo calyptratus*) by Andrews [12]. The occurrence in this species of male heterogamety with homomorphic XY-type sex chromosomes was revealed by the RAD-seq approach, which revealed 13 male-specific RAD-seq markers characterizing the Y chromosome [13]. Subsequently, the use of male-specific RAD-seq markers from *C. calyptratus* in fluorescent in situ hybridization experiments on metaphase spreads of the common chameleon (*C. chamaeleon*) led to the characterization of chromosome 2 (the second largest pair of macrochromosomes) as the sex chromosome pair in the latter species [13,14].

The veiled chameleon is a typical member of the Chamaeleonidae family and is a popular model species in evolutionary developmental biology [15,16] and cytogenetic research [8]. The karyotype of *C. calyptratus* differs from the putative ancestral chameleon karyotype due to its smaller chromosome number and consists of 24 chromosomes with 6 macrochromosome and 6 microchromosome pairs [17]. Multi-tissue transcriptome assemblies of adult and embryonic veiled chameleons are available, which can serve as a resource for evolutionary and developmental biology [18]. However, full genome assemblies of *C. calyptratus* are still not available.

In this study, we identified the major chromosomal rearrangements which occurred in the *C. calyptratus* karyotype by sequencing chromosome-specific flow-sorted DNA libraries and comparing them with the reference iguanian genomes of the green anole (*Anolis carolinensis*), fence lizard (*Sceloporus tristichus*), and desert horned lizard (*Phrynosoma platyrhinos*). We used PCR-assisted mapping of several male-specific RAD-seq markers from Nielsen et al. [13] on a set of flow sorting-derived libraries to identify the sex chromosome pair in the *C. calyptratus* karyotype. In addition, we identified the blocks of the most abundant tandemly arranged repetitive sequences of the veiled chameleon genome and localized them on metaphase chromosomes to reveal possible cytological differences between the X and Y chromosomes.

## 2. Results

### 2.1. FISH with Flow-Sorted Chromosome-Specific Probes of Chamaeleo calyptratus

The flow-sorted karyotype of *C. calyptratus* consisted of 11 well-resolved peaks (Figure 1).

Hybridization of labeled probes obtained by flow sorting on metaphase chromosomes of *C. calyptratus* revealed that 10 of them painted a single chromosome pair each (Figure 2).

Only one probe labeled two microchromosomes (CCA10 and CCA11), as those were co-sorted to the same peak due to the similar sizes and GC content of these microchromosomes.

### 2.2. Sequencing of the Flow-Sorted Chromosome-Specific Probes

All flow sorting-derived chromosome-specific DNA libraries were sequenced on the MiSeq Illumina platform. An estimated 0.24–0.35 million reads were obtained per DNA pool (Appendix A). After processing the resulting reads using the ‘DOPseq_analyzer’ pipeline [19], around 10–20% of them were mapped on targeted genomes.

The obtained reads of the *C. calyptratus* chromosome-specific DNA libraries were compared with the available assembled genomes of *Anolis carolinensis* (AnoCar2.0, Appendix A), *Phrynosoma platyrhinos* (MUOH_PhPlat_1.1, Appendix A), and *Sceloporus tristichus* (ASM1680106v1, Appendix A). Comparative analysis of the genetic content of *C. calyptratus* (CCA) macrochromosomes with the assembled genome of *P. platyrhinos* (PPL) and *A. carolinensis* (ACA) demonstrated a high conservation of syntenic blocks *in toto* for CCA2, CCA4, and CCA6 macrochromosomes, whereas CCA1, CCA3, and CCA5 consisted of 2–3 conserved syntenic blocks (Table 1).

Thus, CCA1 demonstrated a high degree of synteny to the entire macrochromosome PPL2 (=ACA2) and two microchromosomes: a fragment of microchromosome PPL6 and microchromosome PPL8. The fragment CCA1part’/microPPL6part’/microPPL8 is syntenic to the *A. carolinensis* autosomes 15 and 16. CCA3 and CCA5 revealed synteny not only with the entire PPL3 and PPL5 macrochromosomes but also with microchromosomes PPL11 (=ACA18) and PPL9 (=ACAX). Besides, a small fragment of ACA6 synteny was found on CCA3. We detected several lineage-specific chromosomal fusions among some microchromosomes. Thus, CCA7 consists of the homologues of microPPL1 and a fragment of microPPL6part’’ (ACA7 + ACA14), and CCA8 is syntenic to microPPL4 and microPPL10 (ACA9 + ACA17). One of the peaks contains chromosomes CCA10 and CCA11, which probably correspond to microPPL3 and microPPL5 (ACA10 and ACA11).

The results of the alignment of *C. calyptratus* DNA libraries to the *S. tristichus* genome showed inconsistent results. For example, homology to the same fragment of the microchromosome STR8 was consistently found in CCA4, CCA9, and CCA12. It is likely related to the insufficient quality of the genome assembly of *S. tristicus* for such an analysis.

### 2.3. Identification of the Putative Sex Chromosome Pair

Fluorescent in situ hybridization with the male-specific RAD-seq markers as probes did not show any specific signals on the metaphase chromosomes of *C. calyptratus*. PCR-assisted mapping of male-specific RAD-seq markers [13] revealed that only the M2 marker produced a single bold band and indicated CCA5 as the marker-carrying chromosome pair (Figure 3).

Other male-specific RAD-seq markers did not show any chromosome-specific band patterns (Appendix A).

### 2.4. Repetitive DNA Bioinformatic Analysis and FISH Localization

The MGI sequencing produced 13,844,342 reads. Only 6,387,179 randomly selected high-quality reads were analyzed using RepeatExplorer2 and TAREAN software. A total of five putative tandemly arranged repetitive elements were isolated from the genomic DNA sequences. The characteristics of these elements are given in Table 2.

CCA_s2 and CCA_s5 probes labeled a pericentromeric region on CCA1 and co-localized with the ITSs on this chromosome (Figure 4a,d and Appendix A).

The CCA_s1 probe produced signals at the centromeric regions of CCA2 and CCA3, whereas the CCA_s3 probe labeled the centromeric region of CCA4 (Figure 4a,b). The CCA_s4 probe produced interspersed signals on all chromosome pairs, suggesting that it represents a part of a dispersed repetitive element (Appendix A).

An NCBI blast alignment of the obtained monomer sequences of the repetitive elements against the published reference genomes of squamate reptiles revealed non-significant similarities between S2 repeat and a genome fragment of the western terrestrial garter snake *Thamnophis elegans* (NC_045548.1, e-value 0.004) and also between S4 repeat and a genome fragment of the bearded agama *Pogona vitticeps* (NW_018150954.1, e-value 0.002). The alignment of other repetitive element sequences against the published reference genomes of squamate reptiles did not reveal any significant similarities, which indicates that the elements may be Chamaeleonidae-specific.

### 2.5. FISH with Telomeric and rDNA Probes of Chamaeleo calyptratus

Previously, interstitial telomeric sequences (ITSs) were localized on one of the macrochromosomes, and the NORs were localized on the first pair of macrochromosomes [8]. The dual-color FISH with telomeric and rDNA probes revealed both signals on CCA1 (Figure 4c).

## 3. Discussion

The karyotype of the veiled chameleon contains a smaller number of chromosome pairs than the putative ancestral karyotype of Chamaeleonidae [8]. Reduction of chromosome number is not a rare event in the reptilian karyotype evolution and usually occurs either by Robertsonian or tandem fusion. Previously, we described several cases of chromosomal fusions in the genomes of *Sceloporus malachiticus* [20], *Ctenonotus pogus* and *C. sabanus* [21,22], *Norops sagrei* [21,22,23], and *N. valencienni* [21] from the Pleurodonta clade. Most of these homogeneous fusions occurred between microchromosomes and the sex chromosomes, which are also microchromosomes. In the case of *C. calyptratus*, we detected not only homogeneous fusions between microchromosomes but also heterogeneous fusions between macro- and microchromosomes. Usually, micro- and macrochromosomes are different in GC and genic content. A high conservation of microchromosomes (without any fusions with macrochromosomes) was demonstrated in some lineages extending over millions of years [24,25,26,27]. Heterogeneous fusions are rare in vertebrates, as macro- and microchromosomes may have different positions in the nucleus and different transcription activity and replication timing [24,28]. However, the phenomenon of heterogeneous fusions has recently been reported for eagles, alligators, two turtle species [29,30], and several squamate reptiles [31]. Besides, in some taxa, i.e., mammals, microchromosomes disappeared early during evolution through fusions with macro-elements [29].

Previous studies described the chromosome carrying the 45S rDNA genes as CCA1 [8]. However, the genetic content of the corresponding flow sorting library demonstrates a high level of synteny with ACA2/PPL2. Here, we demonstrate that the physical size of CCA1 increased due to the fusion of the homolog of ACA2 with the microchromosomes homologous to ACA15 and ACA16.

The genome of the veiled chameleon is characterized by the accumulation of tandemly repeated DNA in the centromeric and pericentromeric regions of macrochromosomes. The co-localization of two tandemly repeated elements with ITSs on CCA1 shows a complex content of these C-positive interstitial heterochromatic blocks, previously described only by the C-banding technique [8]. Some authors assume that the heterochromatic ITSs at pericentromeric regions could represent the remnants of chromosome rearrangements, such as Robertsonian fusion [32]. ITSs in centromeric and pericentromeric regions are described for a few species of iguanas and agamas [33]; however, the presence of other tandemly arranged repeats has not yet been demonstrated in these species.

Similar karyotypes and relatively shallow divergence time (about 10–13 million years ago) [3] between *C. calyptratus* and *C. chamaeleo* suggest that these species may share the same sex chromosome pair. Previous data reported the second largest chromosome pair as a putative sex chromosome of *C. chamaeleo* [14], whereas here, we identified CCA5 as a putative sex chromosome pair of *C. calyptratus*. The data obtained on *C. chamaeleo* needs further revision and additional evidence to validate the FISH results.

The putative sex chromosomes of *C. calyptratus* are rich in syntenic regions involved in forming sex chromosomes in many reptiles. ACAX (syntenic to chicken chromosome GGA15, and here we demonstrate its homology to CCA5) is a conserved sex chromosome element of all pleurodont iguanas (except Corytophanidae) [21,22,23,34,35]. It also acts as a sex chromosome in soft-shelled turtles (Testudines: Trionychidae) [36,37] and geckos of the genus *Paroedura* [38]. ACA5 (which represents another fragment of CCA5) contains the syntenic block of the chicken chromosome GGA4q, which forms the sex chromosome of the pygopodid geckos [39], the Murray River turtle (*Emydura macquarii*) [40], and a fragment of chicken chromosome 1, which can be a part of the XY of skinks [41] and eublepharids geckos [42].

Genotypic sex determination is widespread among squamate reptiles; however, the master sex-determining gene is yet to be found. The fragment of CCA5 (syntenic to ACA5) contains at least three genes, *PITX2*, *SOX5*, and *SOX10*, which are involved in the gonadal embryonic development of vertebrates and could be candidates for the master sex-determining gene [43,44,45,46].

The putative sex chromosomes of *C. calyptratus* demonstrate the same syntenic group with most pleurodont iguanas (ACAX/GGA15). However, this group has not been revealed in studies of the genetic content of the sex chromosomes of other acrodont lizards. For example, the genetic content of the micro-Z and W chromosomes of the bearded dragon (*Pogona vitticeps)* is highly syntenic with the chicken chromosomes 17 and 23 (ACA16 and a fragment of ACA9, respectively) [47]. Micro-sex chromosomes of the ZZ/ZW type are also described for chameleons of the genus *Furcifer*. According to this review [48], the genetic content of these sex chromosomes is syntenic with the chicken chromosome 4p (ACA11). Macrochromosomes as sex chromosomes in Iguanian lizards were previously described only for the Qinghai toadhead agama *Phrynocephalus vlangalii* (ZW/ZZ) [49]. As the genetic content of these sex chromosomes is unknown, we could not compare them with other acrodont sex chromosomes.

Thus, this study expands our understanding of the karyotypic and sex chromosome evolution of the veiled chameleon. Further comparative analysis of the chromosomal composition of taxa with unimodal karyotypes, such as crocodiles, geckos, and lacertid lizards, and taxa with bimodal karyotypes with smaller chromosomal numbers than putative ancestral karyotypes are needed. This could help to elucidate if heterogeneous chromosomal fusions are common across squamate reptiles. Sequencing data from chromosome-specific DNA libraries may be useful for future chromosome-level genome assembly of these species.

Our study is the first to identify the putative sex chromosome pair of *C. calyptratus* and its content. Further work on the sex chromosome composition of additional acrodont lizards is needed to determine whether ACAX/GGA15 is an ancestral sex chromosome for both pleurodont and acrodont lizards or whether this similarity results from convergence.

## 4. Materials and Methods

### 4.1. Animals

A male specimen of *Chamaeleo calyptratus* was obtained from the pet trade.

### 4.2. Cell Culture and Chromosome Suspension

The primary fibroblast cell culture and chromosome suspensions of *C. calyptratus* were obtained at the Cambridge Resource Centre for Comparative Genomics, Department of Veterinary Medicine, University of Cambridge, Cambridge, UK, using the previously described protocols [50,51]. All cell lines were deposited in the IMCB SB RAS cell bank (‘The general collection of cell cultures’, 0310-2016-0002).

### 4.3. Chromosome Sorting and Probe Preparation

The set of chromosome-specific male DNA libraries of *C. calyptratus* (CCA) was prepared using the high-speed cell sorter (Mo-Flo Cell Sorter, Beckman Coulter, Brea, CA, USA) at the Cambridge Resource Centre for Comparative Genomics, Department of Veterinary Medicine, University of Cambridge, Cambridge, UK, as described previously [52]. The painting probes were made from the degenerate oligonucleotide-primed polymerase chain reaction (DOP–PCR) amplified samples by incorporation of biotin-dUTP (Sigma, St. Louis, MO, USA) and digoxigenin-dUTP (Sigma) in the secondary DOP-PCR [53].

### 4.4. DNA Extraction

Genomic DNA was extracted from fibroblast cells using the GeneJET Genomic DNA Purification Kit (Thermo Scientific, Waltham, MA, USA) according to the manufacturer’s protocol.

### 4.5. Telomeric and rDNA Probes

Plasmid DNA (pHr13), containing partial human 18S, full 5.8S, partial 28S ribosomal units, and two internal spacers, was used to reveal the localization of 45S rDNA genes [54]. A telomeric probe, synthesized with primers (TTAGGG)_5_ and (CCCTAA)_5_, was used to compare the localization of previously described blocks of interstitial telomeric sequences (ITSs) with the sites of 45S rDNA genes [55].

Probes were amplified and labeled using the «FTP-Display» DNA fragmentation kit (DNA-Display, Russia) according to the manufacturing protocol, incorporating digoxigenin-dUTP (Sigma) and biotin-dUTP (Sigma) for plasmid DNA (pHr13) and telomeric DNA matrices, respectively; the 50 μL reaction mixture contained 0.05 mM modified nucleotides and 80 ng DNA matrices.

### 4.6. Probes of Sex-Specific RAD-Seq Markers

PCR-amplification of five previously described male-specific RAD-loci (M2, M3, M11, M12, M13) [13] was performed to identify the sex chromosome pair of *C. calyptratus* using fluorescent in situ hybridization. The PCR amplification parameters were set by Nielsen et al., 2018 [13], and the annealing temperature of primers was increased up to 58 °C. PCR-amplified RAD sequences were labeled by secondary PCR with the incorporation of biotin-dUTP and digoxigenin-dUTP (Sigma).

### 4.7. Male-Specific RAD-Seq Markers Mapping

Chromosome-specific DNA from flow-sorted chromosomes of *C*. *calyptratus* was used as a template for PCR-assisted mapping of CCA Y-linked RAD-seq markers (M2, M3, M11, M12, M13) [13] to identify the sex chromosome pair.

### 4.8. DNA Sequencing

The set of chromosome-specific DNA libraries of *C. calyptratus* was prepared for sequencing using the TruSeq Nano DNA Low Throughput Library Prep (Illumina). Paired-end sequencing was performed on Illumina MiSeq using ReagentKit v2, 600-cycles (Illumina). The next-generation sequencing (NGS) data were deposited in the NCBI SRA database under accession number PRJNA832590.

Genomic DNA was fragmented in microTUBE AFA Fiber Pre-Split tubes on a Covaris S2 device (Covaris, Woburn, MA, USA). The median length of fragments is about 400 bp, corresponding to the manufacturer’s protocol. DNA libraries were prepared from fragmented DNA using the MGIEasy Universal DNA Library Prep Set (MGI, Shenzhen, China) according to the manufacturer’s protocol. DNA-library sequencing was performed on the MGISEQ-2000 instrument using the DNBSEQ-G400RS High-throughput Sequencing Set (FCL PE100) in the SB RAS Genomic Core Facility (ICBFM SB RAS, Novosibirsk, Russia). The NGS data were deposited in the NCBI SRA database under accession number SRR20394369.

### 4.9. Bioinformatic Analysis

To accurately determine the species identity of the sample, the complete mitochondrial DNA sequence was assembled from raw reads (GenBank ID:OP965518) using the GetOrganelle 1.7.6.1 pipeline [56]. Available sequences from the NCBI Nucleotide database of the complete mitochondrial sequences of the *Chamaeleo* sp. and the *Furcifer oustaleti* as an outgroup were used to make an alignment using MUSCLE v5 [57]. The phylogenetic tree was constructed using the maximum likelihood method with Tamura-Nei substitution in the Mega 11 software suite (Appendix A) [58]

Sequencing data were processed using the ‘DOPseq_analyzer’ pipeline, as described previously [19,20]. *Anolis carolinensis* (AnoCar2.0, ACA), *Phrynosoma platyrhinos* (MUOH_PhPlat_1.1, PPL), and *Sceloporus tristichus* (ASM1680106v1, STR) genomes were used for synteny block identification.

Identification of repetitive DNA clusters was carried out using RepeatExplorer2 [59] and TAREAN 2.3.8 tools [60] as described previously [61]. Only tandem repeats predicted by TAREAN were localized in metaphase chromosomes of *C. calyptratus*. The NCBI BLAST database was used to compare consensus sequences with previously described sequences [62].

### 4.10. Probes for Repetitive DNA Monomers

Probes for tandemly arranged repetitive DNAs were synthesized by PCR with primers described in Appendix A. Primers were designed to cover the largest part of the repetitive unit with fragments ranging in size from 38 (CCA_s4) to 469 bp (CCA_s1). PCR amplification was performed as described previously [61]. The annealing temperature was 60 °C for all primer pairs.

### 4.11. Fluorescent in Situ Hybridization

Dual-color FISH experiments on metaphase chromosomes were performed with standard techniques [63]. Biotin-labeled probes were detected using avidin-FITC (Vector Laboratories) and anti-avidin FITC (Vector Laboratories, Inc., Burlingame, CA, USA), whereas digoxigenin-labeled probes were identified with anti-digoxigenin-Cy^TM^3 (Jackson ImmunoResearch Laboratories, Inc., West Grove, PA, USA). The preparations were analyzed using a BX53 microscope (Olympus, Shinjuku, Japan) equipped with a Baumer Optronics CCD camera and VideoTesT2.0 Image Analysis System (Zenit, St. Petersburg, Russia). Images were processed in Corel PaintShop Photo Pro X3 (Corel Corporation).

## Figures and Tables

**Figure 1 ijms-23-15838-f001:**
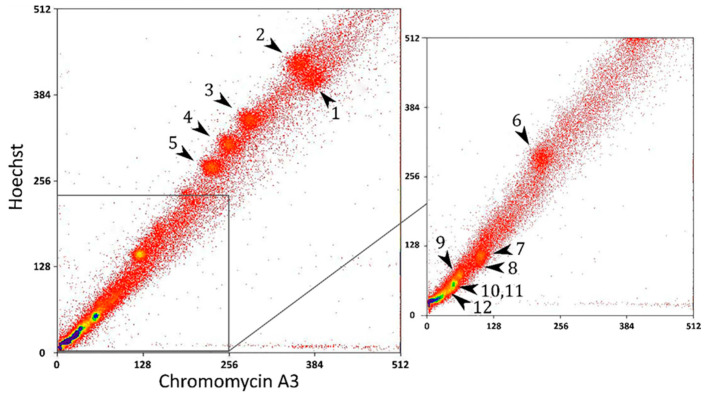
Flow-sorted karyotype of *Chamaeleo calyptratus* (CCA). Chromosome numbers are indicated by the arrowheads.

**Figure 2 ijms-23-15838-f002:**
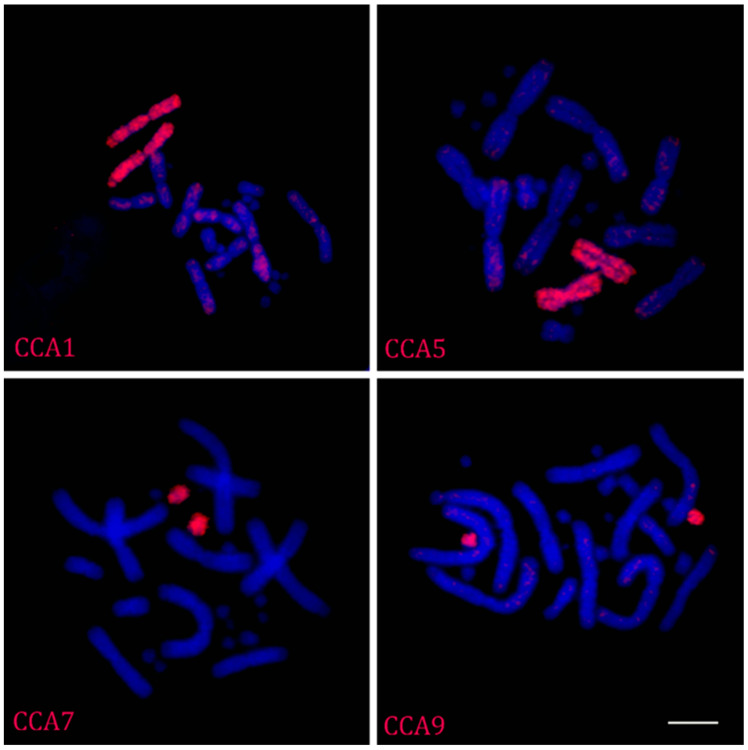
Examples of fluorescent in situ hybridization with the flow-sorted chromosome-specific probes of the *C. calyptratus* on metaphase plates of the same species. Scale bar = 10 µm.

**Figure 3 ijms-23-15838-f003:**
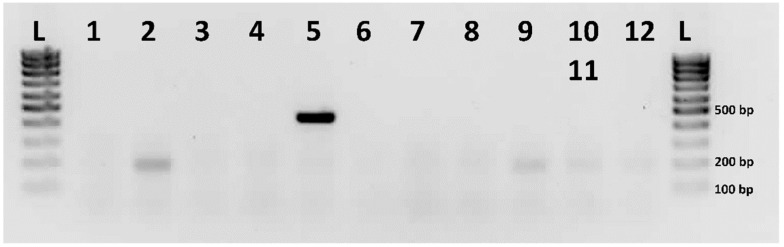
Agarose gel electrophoresis showing the results of PCR-assisted mapping of the male-specific RAD-seq marker M2 in *C. calyptratus*. Numerals refer to the chromosomes, which were identified in the chromosome-specific DNA libraries prepared from *C. calyptratus* flow-sorted chromosomes; L: DNA ladder.

**Figure 4 ijms-23-15838-f004:**
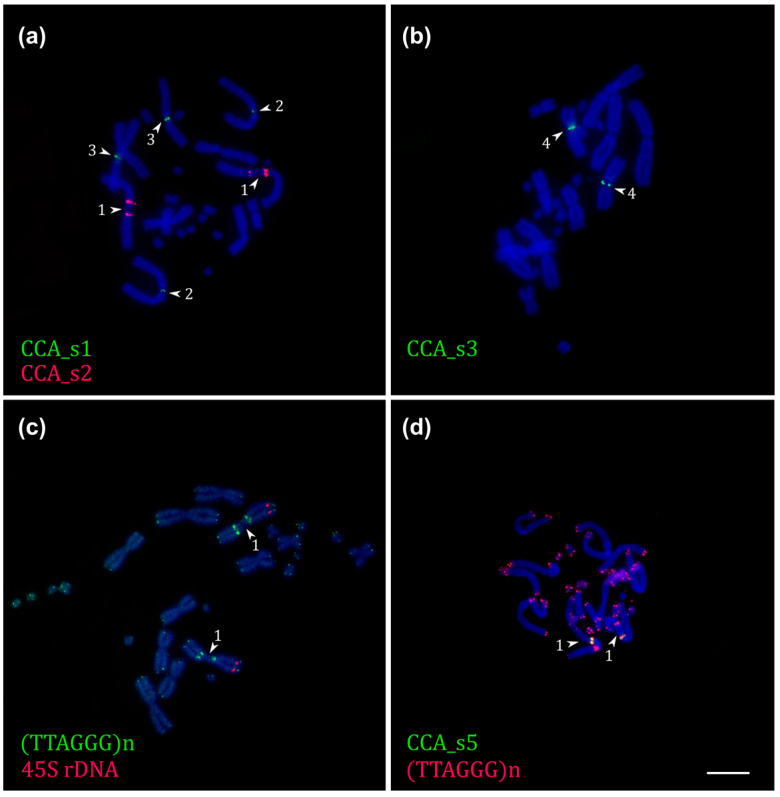
Fluorescent in situ hybridization of *C. calyptratus* repetitive element probes on its metaphase plates. Chromosomes are indicated by the arrowheads. Co-localization of probes are shown: (**a**) CCA_s1 (green) and CCA_s2 (red); (**b**) CCA_s3 (green); (**c**) (TTAGGG)n (green) and 45S rDNA (red); (**d**) CCA_s5 (green) and (TTAGGG)n (red). Scale bar = 10 um.

**Table 1 ijms-23-15838-t001:** The homology between the chromosomes of *C. calyptratus* and reference genomes was inferred from the NGS data.

*C. calyptratus* (CCA) Flow Sorting Peak	*Anolis* *carolinensis* (ACA)	*Phrynosoma platyrhinos* (PPL)	*Sceloporus* *tristichus* (STR)
CCA1	ACA2	PPLma2	STR2
	ACA15	PPLmi6 * (part’) **	STR11
	ACA16	PPLmi8	
CCA2	ACA1	PPLma1	STR1
CCA3	ACA3	PPLma3	STR3
	ACA18	PPLmi11	
	ACA6 (small fragment)		
CCA4	ACA4	PPLma4	STR4
			STR8 (part’’)
CCA5	ACA5 ACAX	PPLma5 PPLmi9 (X)	STR5 STR10
CCA6	ACA6	PPLma6	STR6 (part’)
CCA7	ACA7 ACA14	PPLmi1 PPLmi6 (part’’)	STR6 (part’’) STR9 (part’’)
CCA8	ACA9 ACA17	PPLmi4 PPLmi10	STR7 STR9 (part’) STR9 (part’’’)
CCA9	ACA8	PPLmi2	STR8
CCA10,11	ACA10 ACA11	PPLmi3 PPLmi5	STR6 (part’’’) STR7 (part’)
CCA12	ACA12	PPLmi7	STR8

* “ma” indicates macrochromosomes of *P. platyrhinos* and “mi” indicates microchromosomes. ** “part”–indicates syntenic regions within chromosomes; coordinates of syntenic region boundaries are given in Appendix A.

**Table 2 ijms-23-15838-t002:** Characteristic of the most abundant putative tandemly arranged repeated elements in the veiled chameleon (*C. calyptratus*) genome.

Repetitive Element Name	Monomer Length, bp	Genome Fraction	GC-Content, %	GenBank Accession Number
CCA_s1	462	0.077	35.4	OP297933
CCA_s2	98	0.045	38.8	OP297934
CCA_s3	1228	0.024	39.1	OP297935
CCA_s4	51	0.13	54.9	OP297936
CCA_s5	880	0.021	49	OP297937

## Data Availability

Sequencing data were deposited in the NCBI SRA database under accession no. PRJNA832590. The obtained repetitive element consensus sequences were deposited in the GenBank database (OP297933-OP297937).

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
