# Peer review of "Identification of Iguania Ancestral Syntenic Blocks and Putative Sex Chromosomes in the Veiled Chameleon (Chamaeleo calyptratus, Chamaeleonidae, Iguania)"

_ijms, 2022, doi:10.3390/ijms232415838_

Round 1

Reviewer 1 Report

Tishakova and colleagues carried out a cytogenetic and molecular study in a Chameleon species using different approaches to discuss sex chromosomes in the species and in related groups, bringing an overview regarding the chromosomal rearrangements and its role and relation with the sex chromosomes in the group.

The study is interesting; however, some points need attention.

My main concern is regarding the use of only a single individual and of unknown origin in the study. It is only mentioned that the animal is from pet trade, however, it is interesting to note that in captivity these animals can hybridize with species and genera with 2n different. How did the authors identify the species?

And even if they used COI or some other molecular marker, how can the authors guarantee that in fact the only individual analyzed is a pure species and not a product of interbreeding in captivity? Did the authors deposit the analyzed individual in any scientific collection?

I do not doubt that it is the species in fact, but the use of only a single individual, which does not come from natural environment, makes the support of the conclusions of the study a little weak, especially since it is also a discussion focused on the evolution of sex chromosomes and having analyzed only a single male individual.

It would be very interesting if the authors could also show the gel with the results of mapping the sex-specific markers (that from chromosome 5) in females, as there would be no doubts about the pattern of the RAD-seq marker M2. Again, I do not doubt that in fact the site located on chromosome 5 is a sex-specific marker, but showing the differential amplification pattern of this site would be very welcome in the manuscript.

Results

In figure 4C the authors show the mapping of telomere repeats in green color, which highlights 3 very evident ITS sites in pair 1, and although the other chromosome of the pair 1 presents an overlap, it is still possible to detect such ITSs.

However, in Figure 4D, the ttaggg mapping in green color does not show the same 3 ITSs seen in pair 1, where it is only possible to identify 2 ITSs, one of which is colocalized with the CCA_s5 site. Is it not a metaphase of the same individual and the same species? So, the pattern should be exactly the same. In how many metaphases did the authors perform the mapping of such sequences to confirm the hybridization pattern? Obtaining good chromosomal preparations from reptiles is not easy, but showing better metaphases with no overlap would be more interesting when the intention is to show specific sites and with synteny.

The authors also mention about the synteny of the CCA_s2 marker and the ITSs present in this chromosomal pair. Where are the double fish images of ttaggg and CCA_s2? In the case of the CCA_s5 marker, the authors showed the double fish in par 1, highlighting the synteny with one of the ITSs of this pair, why didn't the authors show the same for the CCA_s2 marker?

Also, pair 1 presents, at least in the 4C image, 3 very evident sites of ITSs, one in the centromeric position, one in the long arm and one in the short arm. My question is: in which of these sites is the ITS synteny with the CCA_s5 site? I say this because the authors mention in line 150 the colocalization of the CCA_s2 and CCA_s5 sites with the ITSs, could the CCA_s2 site be in a position in a specific ITS and the CCA_s5 site in another ITS? Could CCA_s2 and CCA_s5 be located in the same ITS, highlighting synteny not only with the ITS but also between isolated repeat motifs?

This could be better discussed, regarding the role of this synteny in the formation of these ITSs, a possible recruitment of repeat motifs at these sites, which could make them susceptible to other rearrangements or chromosomal breaks, as seen in several species of fish and reptiles.

Regarding the CCA_s4 marker, the authors only state that the pattern found was dispersed in all chromosomes, but again, where are the figures? Authors could at least cite in the text: image/data not shown.

Discussion

In lines 207-209 the authors discuss the possibility that C. calyptratus and C. chamaeleo share the same homomorphic sex pair. And they report that in C. chamaeleo the sexual pair identified as putative sexual was a different pair from the one suggested in the present study in C. calyptratus. Furthermore, they suggest that the cited study (reference 14) needs further analysis and validation by fish.

But, I believe that the present study also needs further investigation in order to validate the pair 5 as the sexual putative. The authors did not map the sex-specific site isolated on the chromosomes of the analyzed species (only 1 male individual). Again, the authors bring an analysis and an evolutionary discussion about sex chromosomes and sequence homologies among lineages, as well as possible chromosomal rearrangements between macro and micro based on just a single male individual, in this sense, the present study also needs further analysis to support the discussions raised.

Reviewer 2 Report

The manuscript by Tishakova et al. aims to study the chromosomal rearrangements in a species of chameleon (Chamaeleo calyptratus). Authors reported fusion and fission events and found conserved syntenies between autosomal and sex chromosomes. The manuscript is well written, data is impressive, and the study is interesting also to a wider audience of cytogenetic community. I recommend this paper for publications in IJMS.

I have only minor comments for authors.

In abstract,  define the abbreviation “CCA5” for first occurrence in the manuscript.

In results, Line 90: “FISH with the labeled probes”.  Which probes?

Line 95, Only one of the probes painted two microchromosomes. Authors should define specifically which probe this corresponds to repeat or other sequence?

In results, Line 99, Resulting reads were processed using the ‘DOPseq_analyzer’ pipe- 99 line [19]. It can be moved to Methods.

Table 1. Line 109; Chromosome numbering should be consistent and throughout the manuscript.

“Chr1” or “1” or specifies related abbreviated names “CCA1”?

Line 111:

* “ma” indicates microchromosomes of P. platyrhinos and “mi” indicates microchromosomes.

Confirm, whether “ma” indicates macrochromosomes or microchromosomes?

Round 2

Reviewer 1 Report

In the current version, the authors addressed most of the suggestions made to improve the manuscript. Only minor revisions along the manuscript are still required in order to amend typos, as for instance in the line 319:

The "acailable" sequences from the NCBI Nucleotide database
